# CTV-FAS: Compensate Texts with Visuals for Generalizable Face Anti-spoofing

## Abstract

Generalizable Face Anti-Spoofing (FAS) approaches have recently gained significant attention for their robustness in unseen scenarios. Recent methods incorporate vision-language models into FAS, capitalizing on their remarkable pre-trained performance to enhance generalization. These methods predominantly rely on text prompts to learn the concept of attacks in FAS. However, certain attacks, such as high-resolution replay attacks, cannot be described linguistically. Relying solely on text prompts cannot accurately tackle such attacks, resulting in performance degradation. To tackle these limitations, we introduce a novel framework named CTV-FAS, designed to exploit visual anchors to compensate for the shortcomings of semantic prompts. Specifically, we employ a Self-Supervised Consistency Module (SSCM) to boost the generalization of visual anchors, which utilizes consistency regularization to facilitate visual feature learning. Subsequently, a Visual Anchors Updating Module (VAUM) is proposed to incorporate the visual anchors through an adaptive updating scheme, guiding the feature learning process from a visual standpoint. Furthermore, we propose an Adaptive Modality Integration Module (AMIM), designed to merge visual and textual information during inference seamlessly. This integration optimizes the synergy between modalities, significantly boosting the efficacy of Face Anti-Spoofing (FAS) tasks. Our extensive experimental evaluations and in-depth analysis affirm that our method outperforms current state-of-the-art counterparts with a notable margin of superiority.

## 1 Introduction

Face recognition techniques have gained significant traction in diverse applications, such as smartphone login, access control, and electronic payments. Nevertheless, face recognition techniques are constantly confronted with a range of potential threats posed by various presentation attacks, such as printed photos Anjos & Marcel (2011), masks Erdogmus & Marcel (2013), and video replays Smith et al. (2015). To mitigate these attacks, researchers propose various Face Anti-Spoofing (FAS) methods that rely on either hand-crafted features Yang et al. (2013); Kim et al. (2012); Zhang et al. (2011); Kim et al. (2013); Singh et al. (2014) or deeply-learned features Zhou et al. (2023); Zhang et al. (2020a); Yu et al. (2021); Wang et al. (2021a; 2023b) for detection.

Although existing methods have shown promising performance in intra-dataset scenarios, they encounter difficulties in effectively generalizing to unseen domains due to the inherent domain gap between the source and target distributions. To address this challenge, domain generalization (DG) methods have been incorporated into FAS tasks to learn domain-agnostic discriminative features from multiple source domains, allowing for better generalization to unseen domains. Adversarial learning-based methods Jia et al. (2020); Liu et al. (2022a); Wang et al. (2022c) and meta-learning-based methods Du et al.; Kim & Lee (2021); Liu et al. (2021b) are commonly used in DG. Despite numerous attempts to enhance the generalization ability, uni-modal models have yet to truly overcome this challenge.

As visual-language methods gain prominence, researchers increasingly explore the use of cross-modal foundation models to bridge the visual domain gap via language modalities. Based on the vision-language pretrain model (*i.e.*, CLIP Radford et al. (2021a)), FLIP Srivatsan et al. (2023) aligns the image representation with an ensemble of coarse-grained class descriptions to improves FAS generalizability in low-data regimes. VL-FAS Fang et al. employs content-related prompts to guide

Figure 1: Comparison of previous VL methods and our proposed CTV-FAS.

the model to focus on specific facial regions. Supervised by the text semantic prompts, these methods indeed achieve remarkable performance in domain generalization settings. However, their limitation stems from an exclusive dependence on semantic prompts for supervisory guidance during learning, neglecting the potential advantages of incorporating visual cues. Because FAS tasks involve specific attack types, such as high-resolution paper and replays, which cannot be described linguistically. This reliance on purely semantic prompts results in sub-optimal generalization performance, as shown in Fig. 1(a).

To address this challenge, we propose a novel framework called CTV-FAS, which adopts visual anchors to compensate for the deficiency of semantic prompts in FAS tasks, as shown in Fig. 1(b). Specifically, CTV-FAS proposes a semantic-visual adaptive ensemble framework to effectively perceive discriminative visual features for FAS tasks with three designs, namely Self-Supervised Consistency Module (SSCM), Visual Anchors Updating Module (VAUM)), and Adaptive Modality Integration Module (AMIM). **What visual cues are robust enough to differentiate between the real person and paper/replay attack?** The proposed Self-Supervised Consistency Module utilizes the self-supervised methods to mine fine-grained features between the global view and local view, thus improving the robustness of the visual anchor representation. **What visual anchors can compensate for the deficiency of semantic prompts?** VAUM is further used to dynamically optimize the visual anchors during training. In principle, visual cues that have the lowest cosine similarity with their corresponding semantic prompts are selected. Moreover, visual features from a momentum teacher model are used for the superiority of stability. During inference, AMIM is introduced to effectively combine the predictions from semantic prompts and visual cues. It enhances the reliability of the fused results by increasing the weights of high-confidence predictions and decreasing the ones of low-confidence, thus fully exploit the advantages of both text and visual anchors to improve generalization ability.

- We present the first attempt of unifying semantic prompts and discriminative visual cues via complementary mechanisms, which is a new insight of CLIP-based model adaption for FAS tasks.
- We develop a strong semantic-visual framework called CTV-FAS equipped with three novel designs, *i.e.*, Self-Supervised Consistency Module (SSCM), Visual Anchors Updating Module (VAUM) and Adaptive Modality Integration Module (AMIM).
- Extensive experiments and analysis demonstrate the superiority of CTV-FAS over state-of-the-art uni-modal and cross-modal methods by a significant margin on OCIM datasets, *e.g.*, +27.07 in I→O setting.

## 2 RELATED WORK

### 2.1 FACE ANTI-SPOOFING

Conventional methods mainly utilize various hand-crafted features such as LBP Chingovska et al. (2012a); Boulkenafet et al. (2015); de Freitas Pereira et al. (2013), HoG Komulainen et al. (2013); Yin et al. (2016); Schwartz et al. (2011), and SIFT Agarwal et al. (2016); Boulkenafet et al. (2016); Patel et al. (2015), to differentiate real and fake faces. However, the performance of these methods is underwhelming due to the shallow structure. With the advent of deep learning, many deep architectures are employed to extract more discriminative features. This evolution included the integration of auxiliary signals like depth maps Shao et al. (2019a), r-ppg signals Niu et al. (2020), or reflection map Yu et al. (2020a) to enhance detection capabilities. Despite advancements in intra-

dataset settings, substantial performance degradation is observed in target domains due to pronounced domain shifts. FAS techniques employ domain adaptation (DA) Zhou et al. (2022b); Li et al. (2018); Wang et al. (2021a); Jia et al. (2021); Panwar et al. (2021) to mitigate the distribution disparities between source and target domains. However, the acquisition of a sufficient volume of unlabeled target data often poses significant challenges and incurs high costs. Domain generalization (DG) methods have been incorporated into FAS tasks to facilitate the learning of domain-agnostic features via adversarial learning Jia et al. (2020); Liu et al. (2022a); Wang et al. (2022c), meta-learning Du et al.; Kim & Lee (2021); Liu et al. (2021b); Zhou et al. (2022a) and instance whitening Zhou et al. (2023), thereby enhancing generalization to unseen domains. Recently, Vision Transformers (ViT)-based approach Liu et al. (2023); Huang et al. (2022); George & Marcel (2021) posits that ViT can discern long-range dependencies for superior generalization. However, relying only on image data can limit its generalization capabilities in unseen domains. The emergence of visual-language methods offers new potential to address the aforementioned issues.

## 2.2 Vision-Language Models

Guided by natural language supervision, vision-language pretraining has recently surfaced as a promising approach for image Chen et al. (2021a); Radford et al. (2021b); Wang et al. (2022b); Li et al. (2022); Zeng et al. (2021) and video understanding Wang et al. (2021b); Wu et al. (2023); Wang et al. (2023a); Cheng et al. (2023); Pramanick et al. (2023). These approaches diverge from the conventional method of utilizing discrete labels, offering a novel paradigm for recognition based on the alignment of visual and text features. It is inherently suited for zero-shot transfer across various downstream tasks Nag et al. (2022); Zheng et al. (2023); Goyal et al. (2023); Shu et al. (2022); Cui et al. (2022). Several studies have investigated the application of the transferable knowledge from pre-trained models to address tasks such as visual question answering (VQA) Parelli et al. (2023); Li et al. (2023b;a), zero-shot object detection Nag et al. (2022); Xie & Zheng (2022); Shu et al. (2022), and image captioning Hu et al. (2022); Fei et al. (2023); Zhong et al. (2022), etc. Recent efforts have sought to leverage visual-language methods to bolster the cross-dataset generalization of FAS tasks Srivatsan et al. (2023); Mu et al. (2023); Fang et al.. These studies posit that text, rich in domain-invariant information, can enhance model generalization. However, these methods rely solely on semantic prompts for supervision, ignoring the potential benefits of visual cues, which leads to unsatisfactory generalization ability. In contrast, we propose a novel framework called CTV-FAS, which explores visual cues to compensate for the shortcomings of semantic prompts in FAS tasks.

## 3 Methodology

### 3.1 Overview

An overview of the proposed CTV-FAS method is depicted in Fig. 2, comprising three main components: SSCM, VAUM, and AMIM. In the training process, SSCM employs varying degrees of data augmentation for self-supervised learning to enhance the model's consistency and robustness. Subsequently, VAUM captures discriminative visual cues to update the visual anchor cache. During inference, AMIM adaptively ensembles the predictions from both semantic and visual branches.

### 3.2 Self-Supervised Consistency Module (SSCM)

To perceive more granular representational details and enhance the model's consistency and robustness with respect to visual cues, we integrate a self-supervised mechanism predicated on patch-masked images (where 75% of the patches are removed from the original images). In practice, both the teacher and student are initialized with the weights of CLIP, sharing the same configurations. The patch-masked images are then input into the student model to get the feature $F_L$, and the global images are sent to the teacher to gain the feature $F_G$. The consistency objective is for the student model to reconstruct the comprehensive, generalized features learned from the teacher model. This process is guided by a cosine loss function, which ensures the alignment of the student's output with the teacher's robust features:

$$\mathcal{L}_{\cos} = 1 - \frac{\mathbf{f}_L \cdot \mathbf{f}_G}{\|\mathbf{f}_L\| \|\mathbf{f}_G\|}. \tag{1}$$

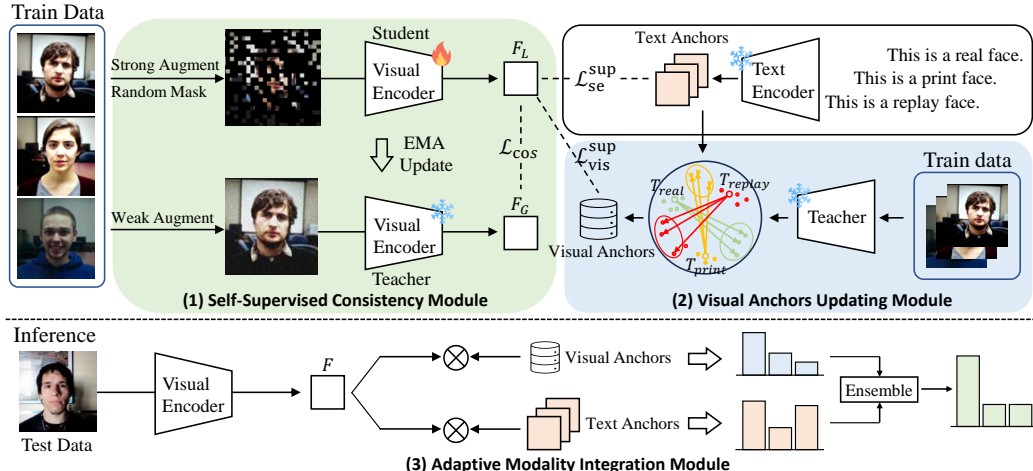

Figure 2: **The overall semantic-visual framework of our proposed CTV-FAS.** CTV-FAS includes three novel designs, namely SSCM, VAUM and AMIM. Different augmented images are passed to SSCM to seek robust and generalizable visual features. Subsequently, visual anchors are optimized to grasp the discriminative visual cues via VAUM, which compensate for semantic prompts. During inference, AMIM is used to ensemble the predictions of two branches adaptively.

In SSCM, the masking of data propels the learning of nuanced features, while the self-supervised methodology amplifies the model's robustness in a teacher-student mutual learning method. The teacher network $\mathcal{T}(\cdot)$ is frozen during training and is updated via an exponential moving average (EMA Tarvainen & Valpola (2017)) predicated on the current model's parameters. This process is articulated as follows:

$$\theta_t^{(t+1)} = \gamma\theta_t^{(t)} + (1 - \gamma)\theta_s^{(t)}, \tag{2}$$

where $\theta_t$ and $\theta_s$ represent the parameters of the teacher and student model, respectively, at training step $t$, and $\gamma$ is the decay rate controlling the update momentum.

## 3.3 VISUAL ANCHORS UPDATING MODULE

In FAS tasks, there are some specific attack types, such as high-resolution replay attacks, that cannot be described with semantic class descriptions. Merely using semantic prompts is insufficient to accurately perceive such attacks, meanwhile destroying the generalization of pre-trained models. To address this challenge, we introduce visual anchors that are specifically designed to compensate for the limitation of semantic prompts. The optimization of visual anchors is a key component of our CTV-FAS. We dynamically update visual anchors during the training process to serve as another anchor for the model. To ensure the robustness and stability of these visual anchors, we employ visual features generated by the teacher network $\mathcal{T}(\cdot)$, built with a momentum visual encoder, to update the cache.

To address the limitations of semantic prompts, we prioritize enhancing visual anchors by incorporating visual cues that are hard for semantic prompts to detect. Therefore, visual cues that exhibit low cosine similarity to their associated semantic prompts are identified as samples that are difficult for semantic prompts to perceive. The cosine similarity between two vectors $\mathbf{a}$ and $\mathbf{b}$ is defined as:

$$\cos(\mathbf{a}, \mathbf{b}) = \frac{\mathbf{a} \cdot \mathbf{b}}{\|\mathbf{a}\|\|\mathbf{b}\|}, \tag{3}$$

where $\cdot$ denotes the dot product and $\|\cdot\|$ denotes the vector norm. The update mechanism for the visual anchor embedding is then given by:

$$\mathbf{P}_v^{(t+1)} = \beta\mathbf{P}_v^{(t)} + (1 - \beta)\mathcal{T}(\mathbf{I})_t. \tag{4}$$

where $\mathbf{P}_v^{(t)}$ represents the visual anchor embedding at updating step $t$, $\beta \in [0, 1]$ is the momentum coefficient, $\mathbf{I}$ is the selected hard images. To enhance training stability, we update the visual anchor once per epoch by scanning the entire dataset. This selective updating strategy ensures that the visual anchors are refined with features that are poorly represented by semantic prompts.

## 3.4 ADAPTIVE MODALITY INTEGRATION MODULE

During inference, AMIM is adopted to ensemble the predictions of semantic and visual anchors. In theory, when the entropy of the model's predictions is high, the model is in a state of uncertainty regarding the data, which increases the likelihood of misclassification. Semantic prompts may not respond effectively to attack categories that cannot be well-described semantically, often resulting in predictions with higher entropy. The design of visual anchors compensates for this deficiency in semantic prompts. Furthermore, we introduce an adaptive ensemble method where the entropy of the model's probability distribution dictates the ensemble weights for the predictions, ensuring a more reliable and accurate decision-making process. The entropy of the semantic and visual predictions are $H(\mathbf{q}_s)$ and $H(\mathbf{q}_v)$, which are calculated as:

$$H(\mathbf{q}_s) = -\sum_i q_{s,i} \log(q_{s,i}); \quad H(\mathbf{q}_v) = -\sum_i q_{v,i} \log(q_{v,i}). \tag{5}$$

where $q_{s,i}$ and $q_{v,i}$ are the predicted probability of class $i$ by the semantic and visual branch respectively. The fusion weight for the semantic branch and visual branch is $w_s$ and $w_v$ respectively, which are inversely related to its entropy and are scaled by a power function to further emphasize lower-entropy predictions:

$$w_s = \left(1 - \frac{H(\mathbf{q}_s)}{H_{\max}}\right)^\alpha; \quad w_v = \left(1 - \frac{H(\mathbf{q}_v)}{H_{\max}}\right)^\alpha. \tag{6}$$

where $H_{\max}$ is the maximum possible entropy, indicating complete uncertainty, and $\alpha > 1$ is a scaling exponent that increases the weight of lower-entropy predictions. The final fused prediction, $\mathbf{q}_f$, is computed by combining the weighted predictions from both the semantic and visual branches:

$$\mathbf{q}_f = \frac{w_s}{w_s + w_v}\mathbf{q}_s + \frac{w_v}{w_s + w_v}\mathbf{q}_v. \tag{7}$$

AMIM ensures that the ensemble metric leverages the strengths of both semantic prompts and visual anchors, dynamically adjusting their contributions based on the certainty of their predictions.

## 3.5 OVERALL TRAINING AND OPTIMIZATION

The framework of this paper is built upon the CLIP. The training objective of CLIP is to maximize the cosine similarity $sim(\cdot, \cdot)$ of the paired image and semantic prompt embedding $P_s$ while minimizing the cosine similarity of the unpaired ones. The image embedding $V$ is extracted by an image encoder $E_v(\cdot)$ and semantic prompt embedding $P_s$ is gained by a text encoder $E_t(\cdot)$.

$$P_s = E_t(T_k); \quad V = E_v(I). \tag{8}$$

where $T_K$ is the sentence describing the K categories. We employ cross-entropy loss to bring matching pairs closer and separate non-matching pairs in feature space, and thus the loss for the anchor is defined as:

$$L_{ce}(x, y, P) = -\frac{1}{N}\sum_{i=1}^{N} y_i \log(sim(V_{x_i}, P_{y_i})), \tag{9}$$

$$with \; sim(V_{x_i}, P_{y_i}) = V_{x_i}^{\mathrm{T}} P_{y_i}/\|V_{x_i}\|\|P_{y_i}\|.$$

The proposed framework has an additional visual branch compared to the CLIP, thus necessitating the calculation of the cross-entropy for the visual branch. The overall cross-entropy loss is:

$$\begin{aligned} L_{ce}(x, y) &= L_{se}(x, y, P_s) + L_{vis}(x, y, P_v) \\ &= L_{ce}(x, y, P_s) + L_{ce}(x, y, P_v). \end{aligned} \tag{10}$$

To further enhance the model's robustness against data variations, we fellow FLIP to employ a SimCLR loss for auxiliary training. This approach generates two views ($I_{v_1}$ and $I_{v_2}$) of a given image $I$ through distinct transformations. The features of the two transformed images are extracted by the image encoder $E_v$ and subsequently projected via a non-linear projection network $\mathcal{H}$. A contrastive loss is then applied to the projected features. $f_{v_1} = \mathrm{E}_v(I_{v_1}), f_{v_2} = \mathrm{E}_v(I_{v_2})$. $h_1 = \mathcal{H}(f_{v_1}), h_2 = \mathcal{H}(f_{v_2}), h_1, h_2 \in \mathbb{R}^{d_h}$.

$$L_{simCLR} = simCLR(h_1, h_2)$$

. Overall, we formulate the joint optimization objective as:

$$L = L_{ce} + \lambda_1 L_{cos} + \lambda_2 L_{simCLR} \tag{11}$$

where $\lambda_1$ and $\lambda_2$ is hyper-parameters.

Table 1: Evaluation of cross-domain performance in Protocol 1, between MSU-MFSD (**M**), CASIA-MFSD (**C**), Replay Attack (**I**) and OULU-NPU (**O**) with the assessment metrics being HTER and AUC. The * indicates using the CelebA-Spoof [83] as the supplementary source dataset.

| Method | OCI → M | | OMI → C | | OCM → I | | ICM → O | | Avg. |
|---|---|---|---|---|---|---|---|---|---|
| | HTER | AUC | HTER | AUC | HTER | AUC | HTER | AUC | HTER |
| MADDG (CVPR' 19) Shao et al. (2019b) | 17.69 | 88.06 | 24.50 | 84.51 | 22.19 | 84.99 | 27.98 | 80.02 | 23.09 |
| MDDR (CVPR' 20) Wang et al. (2020a) | 17.02 | 90.10 | 19.68 | 87.43 | 20.87 | 86.72 | 25.02 | 81.47 | 20.64 |
| NAS-FAS (TPAMI' 20) Yu et al. (2020b) | 16.85 | 90.42 | 15.21 | 92.64 | 11.63 | 96.98 | 13.16 | 94.18 | 14.21 |
| RFMeta (AAAI' 20) Shao et al. (2020) | 13.89 | 93.98 | 20.27 | 88.16 | 17.30 | 90.48 | 16.45 | 91.16 | 16.97 |
| $D^2$AM (AAAI' 21) Chen et al. (2021b) | 12.70 | 95.66 | 20.98 | 85.58 | 15.43 | 91.22 | 15.27 | 90.87 | 16.09 |
| DRDG (IJCAI' 21) Liu et al. (2021c) | 12.43 | 95.81 | 19.05 | 88.79 | 15.56 | 91.79 | 15.63 | 91.75 | 15.66 |
| Self-DA (AAAI' 21) Wang et al. (2021a) | 15.40 | 91.80 | 24.50 | 84.40 | 15.60 | 90.10 | 23.10 | 84.30 | 19.65 |
| ANRL (ACM MM' 21) Liu et al. (2021b) | 10.83 | 96.75 | 17.85 | 89.26 | 16.03 | 91.04 | 15.67 | 91.90 | 15.09 |
| FGHV (AAAI' 21) Liu et al. (2022b) | 9.17 | 96.92 | 12.47 | 93.47 | 16.29 | 90.11 | 13.58 | 93.55 | 12.87 |
| SSDG-R (CVPR' 20) Jia et al. (2020) | 7.38 | 97.17 | 10.44 | 95.94 | 11.71 | 96.59 | 15.61 | 91.54 | 11.28 |
| SSAN-R (CVPR' 22) Wang et al. (2022c) | 6.67 | 98.75 | 10.00 | 96.67 | 8.88 | 96.79 | 13.72 | 93.63 | 9.80 |
| PatchNet (CVPR' 22) Wang et al. (2022a) | 7.10 | 98.46 | 11.33 | 94.58 | 13.40 | 95.67 | 11.82 | 95.07 | 10.90 |
| GDA (ECCV' 22) Zhou et al. (2022b) | 9.20 | 98.00 | 12.20 | 93.00 | 10.00 | 96.00 | 14.40 | 92.60 | 11.45 |
| AMEL (ACM MM' 22) Zhou et al. (2022a) | 10.23 | 96.62 | 11.88 | 94.39 | 18.60 | 88.79 | 11.31 | 93.96 | 13.00 |
| IADG (CVPR' 23) Zhou et al. (2023) | 5.41 | 98.19 | 8.70 | 96.44 | 10.62 | 94.50 | 8.86 | 97.14 | 8.40 |
| GAC-FAS (CVPR' 24) Le & Woo (2024) | 5.00 | 97.56 | 8.20 | 95.16 | 4.29 | 98.87 | 8.60 | 97.16 | 6.52 |
| DiVT-M (WACV' 23) Liao et al. (2023) | 2.86 | 99.14 | 8.67 | 96.62 | 3.71 | 99.29 | 13.06 | 94.04 | 7.07 |
| VL-FAS (ICASSP' 24) Fang et al. | 3.13 | 99.31 | 4.00 | 98.64 | 5.00 | 98.90 | 7.92 | 97.05 | 5.01 |
| CTV-FAS (Ours) | **0.92** | **99.96** | **1.8** | **99.45** | **2.65** | **99.6** | **2.11** | **99.66** | **1.87** |
| ViT* (ECCV' 22) Huang et al. (2022) | 1.58 | 99.68 | 5.70 | 98.91 | 9.25 | 97.15 | 7.47 | 98.42 | 6.00 |
| FLIP-MCL* (ICCV' 23) Srivatsan et al. (2023) | 4.95 | 98.11 | **0.54** | **99.98** | 4.25 | 99.07 | 2.31 | 99.63 | 3.01 |
| CTV-FAS* (Ours) | **0.13** | **99.98** | 0.76 | 99.96 | **1.94** | **99.72** | **0.77** | **99.97** | **0.90** |

Table 2: Evaluation of cross-domain performance in Protocol 2, between CASIA-SURF (**S**), CASIA-CeFA (**C**), and WMCA (**W**) with the assessment metrics being HTER and AUC.

| Method | CS → W | | SW → C | | CW → S | | Avg. |
|---|---|---|---|---|---|---|---|
| | HTER | AUC | HTER | AUC | HTER | AUC | HTER |
| ViT (ECCV' 22) Huang et al. (2022) | 7.98 | 97.97 | 11.13 | 95.46 | 13.35 | 94.13 | 10.82 |
| FLIP-MCL (ICCV' 23) Srivatsan et al. (2023) | **4.46** | **99.16** | 9.66 | 96.69 | 11.71 | 95.21 | 8.61 |
| CTV-FAS | 6.7 | 97.39 | **0.95** | **99.93** | **10.37** | **96.24** | **6.12** |

Table 3: Evaluation of cross-domain performance in Protocol 3, for all the 12 different combinations between MSU-MFSD (**M**), CASIA-MFSD (**C**), Replay Attack (**I**) and OULU-NPU (**O**) with the assessment metrics being HTER. The * indicates using the CelebA-Spoof [83] as the supplementary source dataset.

| Method | C→I | C→M | C→O | I→C | I→M | I→O | M→C | M→I | M→O | O→C | O→I | O→M | Avg. |
|---|---|---|---|---|---|---|---|---|---|---|---|---|---|
| ADDA (CVPR' 17) Tzeng et al. (2017) | 41.8 | 36.6 | - | 49.8 | 35.1 | - | 39.0 | 35.2 | - | - | - | - | 39.6 |
| DRCN (ECCV' 16) Ghifary et al. (2016) | 44.4 | 27.6 | - | 48.9 | 42.0 | - | 28.9 | 36.8 | - | - | - | - | 38.1 |
| DupGAN (CVPR' 18) Hu et al. (2018) | 42.4 | 33.4 | - | 46.5 | 36.2 | - | 27.1 | 35.4 | - | - | - | - | 36.8 |
| KSA (TIFS' 18) Li et al. (2018) | 39.3 | 15.1 | - | 12.3 | 33.3 | - | 9.1 | 34.9 | - | - | - | - | 24.0 |
| DR-UDA (TIFS' 20) Wang et al. (2020b) | 15.6 | 9.0 | 28.7 | 34.2 | 29.0 | 38.5 | 16.8 | 3.0 | 30.2 | 19.5 | 25.4 | 27.4 | 23.1 |
| MDDR (CVPR' 20) Wang et al. (2020a) | 26.1 | 20.2 | 24.7 | 39.2 | 23.2 | 33.6 | 34.3 | 8.7 | 31.7 | 21.8 | 27.6 | 22.0 | 26.1 |
| ADA (ICB' 19) Wang et al. (2019) | 17.5 | 9.3 | 29.1 | 41.5 | 30.5 | 39.6 | 17.7 | 5.1 | 31.2 | 19.8 | 26.8 | 31.5 | 25.0 |
| USDAN-Un (PR' 21) Jia et al. (2021) | 16.0 | 9.2 | - | 30.2 | 25.8 | - | 13.3 | 3.4 | - | - | - | - | 16.3 |
| GDA (ECCV' 22) Zhou et al. (2022b) | 15.10 | 5.8 | - | 29.7 | 20.8 | - | 12.2 | 2.5 | - | - | - | - | 14.4 |
| CDFTN-L (AAAI' 23) Yue et al. (2022) | **1.7** | 8.1 | 29.9 | 11.9 | 9.6 | 29.9 | 8.8 | **1.3** | 25.6 | 19.1 | 5.8 | 6.3 | 13.2 |
| CTV-FAS | 11.64 | **1.72** | **2.57** | **2.79** | **1.72** | **2.83** | **1.34** | 2.31 | **3.21** | **0.99** | **5.71** | **1.72** | **3.21** |
| FLIP-MCL* (ICCV' 23) Srivatsan et al. (2023) | 10.57 | 7.15 | 3.91 | **0.68** | 7.22 | 4.22 | **0.19** | 5.88 | 3.95 | 0.19 | 5.69 | 8.40 | 4.84 |
| CTV-FAS* | **4.85** | **1.04** | **1.02** | 0.69 | **0.67** | **1.55** | 0.35 | **1.53** | **1.76** | **0.06** | **4.1** | **0.8** | **1.54** |

Table 4: Ablation studies on each proposed component

| Baseline | VAUM | SSCM | AMIM | C → I | | C → M | | C → O | | Avg. |
|---|---|---|---|---|---|---|---|---|---|---|
| | | | | HTER | AUC | HTER | AUC | HTER | AUC | HTER |
| ✓ | | | | 16.94 | 89.91 | 5.97 | 97.95 | 8.32 | 96.80 | 10.41 |
| ✓ | ✓ | | | 14.22 | 90.15 | 4.11 | 98.78 | 5.44 | 98.72 | 7.92 |
| ✓ | ✓ | ✓ | | 13.43 | 91.33 | 3.32 | 99.35 | 3.87 | 99.34 | 6.87 |
| ✓ | ✓ | | ✓ | 13.46 | 91.58 | 3.19 | 99.57 | 3.76 | 99.12 | 6.80 |
| ✓ | ✓ | ✓ | ✓ | **11.64** | **92.03** | **1.72** | **99.27** | **2.57** | **99.73** | **5.31** |

## 4 EXPERIMENT

### 4.1 EXPERIMENTAL SETTING

**Datasets and DG Protocols.** Our evaluation encompasses two protocols. Strictly following the Huang et al. (2022), we adopt a leave-one-domain-out approach for the two protocols, treating each dataset as a distinct domain to gauge cross-domain capabilities on the remaining domain. **Protocol 1** tests our method on established cross-domain FAS benchmarks: MSU-MFSD (**M**) Wen et al. (2015), CASIA-MFSD (**C**) Zhang et al. (2012), Idiap Replay Attack (**I**) Chingovska et al. (2012b), and OULU-NPU (**O**) Boulkenafet et al. (2017), with scenarios like **OCI → M** indicating **O**, **C**, and **I** as sources and **M** as the target. **Protocols 2** evaluates large-scale Face Anti-Spoofing (FAS) datasets: CASIA-SURF (**S**) Zhang et al. (2020b), CASIA-CeFA (**C**) Liu et al. (2021a), and WMCA (**W**) George et al. (2020), where **CS → W** means **C** and **S** are sources, and **W** is the target. **Protocol 3**, strictly following Yue et al. (2022), is a single-source-to-single-target setup using **M**, **C**, **I**, and **O** datasets, yielding 12 scenarios. To fairly compare with FLIP, we also conduct the above experiments with the auxiliary dataset the CelebA-Spoof. In addition, to better simulate the real-world scenarios without large pre-trained datasets, we also conduct the experiments without CelebA-Spoof.

**Implementation Details.** The image encoder and the text encoder are the dual-stream CLIP where the image encoder adopts the ViT-B/16 structure. Face images are preprocessed to a resolution of $224 \times 224 \times 3$ and segmented into patches measuring $16 \times 16$. The maximum length of the textual token sequence $L$ is set to 77. Our method is implemented with PyTorch and trained with Adam optimizer, with both the learning rate and weight decay initialized at $10^{-6}$. During training, batch sizes are set to 3. For testing, the batch size is set to 10 across all protocols. Each variant of our model undergoes training for a total of 6000 iterations. $\lambda_1$ and $\lambda_2$ are set to 1. The text encoder is frozen and only the image encoder and the parameters of the category prompt are trained.

**Evaluation Metrics.** Following Huang et al. (2022), we assess our model's performance using two metrics: the Half Total Error Rate (HTER) and the Area Under the Receiver Operating Characteristic Curve (AUC). HTER is the average of the False Acceptance and False Rejection Rates, indicating the model's error balance. A lower HTER signifies better performance. AUC measures the model's discrimination capacity, with higher values closer to 1 indicating superior performance and a value of 0.5 suggesting no discriminative ability beyond random chance. These metrics together provide a nuanced picture of the model's effectiveness.

### 4.2 CROSS-DOMAIN FAS PERFORMANCE

The MCIO dataset, being smaller compared to CelebA-Spoof, benefits significantly from the addition of it in bridging the domain gap between different domains. To comprehensively investigate the impact of the proposed method on domain generalization, all protocols were conducted both with and without CelebA-Spoof. Tab. 1, Tab. 2 and Tab. 3 detail the zero-shot cross-domain performance under **Protocols 1-3**, respectively. The results and analyses are as follows.

**Cross-domain performance in Protocol 1.** The proposed framework attained optimal performance, compared to the current state-of-the-art (SOTA) methods, in all settings without CelebA-Spoof (M =+1.94, C=+2.2, I=+1.06, O=+5.81), with an average performance increase of +3.14. With the inclusion of celeb, optimal performance was achieved in three-quarters of the settings (M=+1.45, I=+2.31, O=+1.54), yielding an average enhancement of +2.11. This demonstrates that the supplementation

Table 5: Ablation studies on SSCM

| SW Aug | TS Learning | EMA | C → I | | C → M | | C → O | | Avg. |
|---|---|---|---|---|---|---|---|---|---|
| | | | HTER | AUC | HTER | AUC | HTER | AUC | HTER |
| | | | 13.46 | 91.58 | 3.19 | 99.57 | 3.76 | 99.12 | 6.80 |
| ✓ | | | 12.24 | 91.47 | 1.60 | 99.86 | 3.36 | 99.41 | 5.73 |
| ✓ | | ✓ | 11.9 | 91.87 | 1.90 | 99.25 | 3.81 | 99.24 | 5.87 |
| ✓ | ✓ | | 16.94 | 87 | 4.49 | 98.82 | 8.13 | 97.16 | 9.85 |
| ✓ | ✓ | ✓ | **11.64** | **92.03** | **1.72** | **99.27** | **2.57** | **99.73** | **5.31** |

Table 6: Effects of the function for self-supervised learning.

| Function | C → I | | C → M | | C → O | | Avg. |
|---|---|---|---|---|---|---|---|
| | HTER | AUC | HTER | AUC | HTER | AUC | HTER |
| MSE | 14.18 | 90.23 | 3.20 | 99.43 | 4.02 | 99.15 | 7.13 |
| KL | 12.05 | 95.22 | 2.52 | 98.99 | 3.84 | 99.36 | 6.14 |
| COS | **11.64** | **92.03** | **1.72** | **99.27** | **2.57** | **99.73** | **5.31** |

and proper integration of visual anchors can effectively improve the generalization performance of spoofing detection.

**Cross-domain performance in Protocol 2.** We strictly follow FLIP to further evaluate CTV-FAS on **Protocols 2**, across large-scale Face Anti-Spoofing (FAS) datasets. The experimental results are shown in Tab. 2. We find that our proposed method surpassed the state-of-the-art (SOTA) performance in **SW → C** and **CW → S** settings by +8.71 and +1.34 in terms of Half Total Error Rate (HTER), respectively. This result further validates the effectiveness of the proposed method on large datasets.

**Cross-domain performance in Protocol 3.** In single-source to single-target settings, the proposed CTV-FAS framework surpasses current SOTA methods by a considerable margin of +9.99 and +3.3 in terms of average HTER without and with the inclusion of CelebA-Spoof, respectively. Specifically, for the target domain O, there are substantial improvements of +22.13, +27.07, and +22.39 when selecting C, I, and M as the source domains, respectively, without CelebA-Spoof. When including CelebA-Spoof, in comparison to FLIP-MCL, the proposed method achieves a maximum increase of +7.6 **O → M**. These results confirm that CTV-FAS is capable of learning robust generalizable features and adapting to navigating challenges posed by limited data and domain gaps.

## 4.3 ABLATION STUDIES

Due to the significant domain gap between dataset C and other datasets, transferring knowledge learned from source domain C to other domains results in a considerable performance drop. Furthermore, incorporating CelebA-Spoof as supplementary data for the source domain helps to bridge the gap between the source and target domains. Therefore, to convincingly demonstrate the feasibility of the proposed method for domain generalization, all ablation experiments are conducted in the settings of **C→I**, **C→M**, and **C→O** without using CelebA-Spoof as additional source domain data.

**Effects of the proposed modules.** To explore the impact of each proposed module on the generalization of FAS, we conducted ablation experiments on the proposed modules, using a dual-stream CLIP structure as the baseline. As demonstrated in Tab. 4, incorporating the VAUM module led to +2.49 enhancement in the average HTER, suggesting that visual anchors can effectively compensate for the deficiencies of text prompts in perceiving attack categories that are indescribable through language. The addition of the SSCM module led to +1.05 increase in average HTER, suggesting that SSCM, through self-supervision with patch-masked data augmentation, compels the model to focus on fine-grained features, enhancing generalizability. In this AMIM module, fusion of predictions from two modalities is achieved using the entropy principle, further enhancing their complementarity and leading to a +1.07 improvement in average HTER. Compared to the baseline, the proposed module shows a significant improvement, achieving a +5.1 increase in average HTER.

Table 7: Performance of text and visual branches of CTV-FAS.

| Function | C → I | | C → M | | C → O | | Avg. |
|---|---|---|---|---|---|---|---|
| | HTER | AUC | HTER | AUC | HTER | AUC | HTER |
| CTV-FAS-T | 13.54 | 90.88 | 2.75 | 99.07 | 3.89 | 99.15 | 6.73 |
| CTV-FAS-V | 12.37 | 91.25 | 3.43 | 99.16 | 2.71 | 99.45 | 6.17 |
| CTV-FAS | **11.64** | **92.03** | **1.72** | **99.27** | **2.57** | **99.73** | **5.31** |

Table 8: Comparison of AMIM and common weighting methods.

| Function | C → I | | C → M | | C → O | | Avg. |
|---|---|---|---|---|---|---|---|
| | HTER | AUC | HTER | AUC | HTER | AUC | HTER |
| Mean weighting | 13.43 | 91.33 | 3.32 | 99.35 | 3.87 | 99.34 | 6.87 |
| Confidence weighting | 12.05 | 91.56 | 1.98 | 99.26 | 2.72 | 99.39 | 5.58 |
| AMIM | **11.64** | **92.03** | **1.72** | **99.27** | **2.57** | **99.73** | **5.31** |

**Ablation studies on SSCM.** The ablation results for SSCM in Tab. 5 emphasize the contribution of each design. Strong-weak data augmentations (SW Aug), with the special patch-masked strategy, can improve the robustness of visual features, improving average HTER by +1.07. The teacher-student training (TS Learning) helps provide stable, optimal features and mitigates error accumulation, with a +0.42 improvement compared to applying strong-weak augmentations to a single visual encoder. Additionally, we compared different teacher model update methods. Freezing the teacher model (w.o. EMA) prevented effective guidance, leading to a performance drop of 3.98. In contrast, updating only the student via EMA resulted in a smaller 0.56 decrease. These results confirm the importance of updating the teacher model for optimal performance.

**Effects of different function for self-supervised learning.** Tab. 6 presents the different functions for self-supervised learning. The results show that the cosine loss (COS) function performs the best (+1.82 increase in average HTER compared to MSE loss), while the mean squared error (MSE) loss function performs the worst, with the Kullback-Leibler (KL) divergence in the middle. This indicates that the cosine loss function is the most suitable for self-supervised feature regularization. We observed that after feature normalization, the loss value using the MSE loss function is almost zero, rendering it ineffective. Although the KL divergence can shape the predicted distribution, its performance in feature regularization for anti-spoofing tasks is not as good as that of the cosine loss function.

**Performance of text and visual branches of CTV-FAS.** The performance comparison of text and visual branches in CTV-FAS shown in Tab. 7 that the individual branches, CTV-FAS-T and CTV-FAS-V, exhibit similar performance levels. However, when combined (CTV-FAS), the system achieves enhanced results, such as an improvement of 0.86 in the average HTER compared to CTV-FAS-V alone. This demonstrates the complementary nature of the two branches, leading to a more robust and accurate model.

**Comparison of AMIM and common weighting methods.** The comparison in Tab. 8 demonstrates that the proposed AMIM method outperforms both mean-weighted Ming & Li (2024) and confidence-weighted Sun et al. (2023) ensemble approaches. With the lowest average HTER (5.31) and consistently strong performance across all metrics, AMIM proves its superiority, offering more reliable results compared to commonly used ensemble methods.

**Analysis of ensemble results examples.** Fig. 3(a) showcases several instances of ensemble outcomes, illustrating how the adoption of an adaptive ensemble strategy can successfully leverage visual anchors to rectify inaccuracies in text semantic prompts. This visualization further supports the notion that intelligent weight scaling within an ensemble framework can lead to more accurate and reliable model performance.

**T-SNE visualization of image feature distributions.** In order to clearly understand how CTV-FAS models live data and learn common knowledge across different datasets, we utilize t-SNE to visualize the feature distributions of each domain. Fig. 3(b-c) shows the visualization result, and we can observe that, compared to the FLIP model, our proposed method is able to learn clear

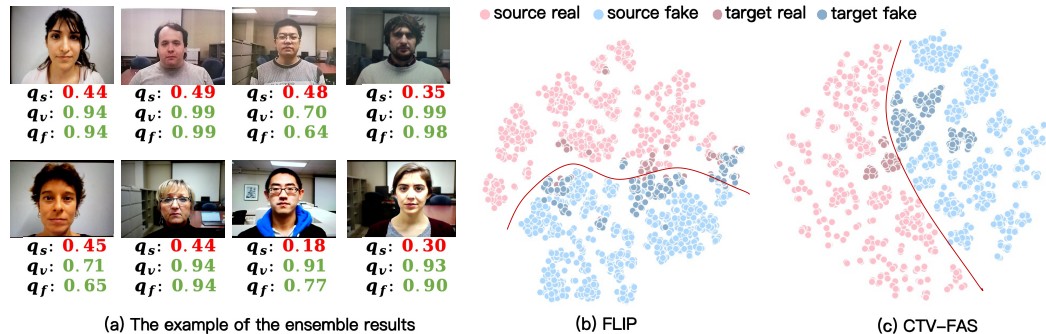

Figure 3: Visualization of the results. (a) is the example of the ensemble results where the first row is for print attack, second row is for replay attack. (b) and (c) are the t-SNE Visualization of FLIP and CTV-FAS respectively.

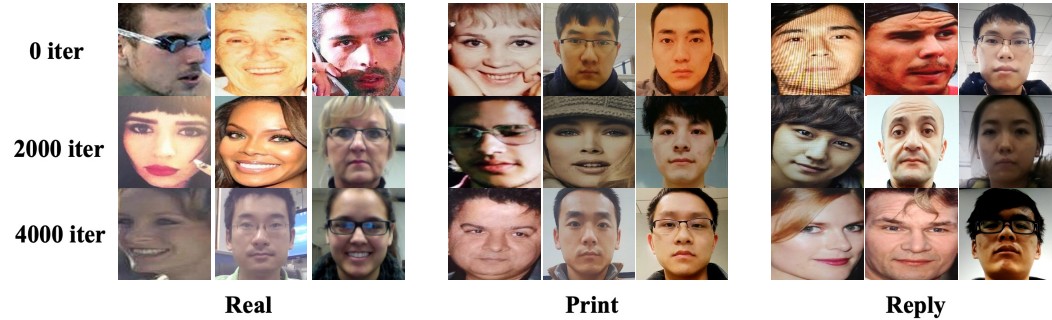

Figure 4: Visualization of the visual anchors of different classes in different training steps.

segmentation boundaries on the source dataset, indicating the effectiveness of SSCM in modeling image distributions. Furthermore, on the target dataset that has not been trained on, our method is also capable of learning clear decision boundaries, and the distributions of the source and target datasets are similar. This demonstrates that through learning with the SSCM module, our model acquires features that exhibit enhanced robustness across domains.

**Visualization of the visual anchors.** Figure 4 illustrates the progression of visual anchors across different training iterations. As training progresses, the selected anchors become increasingly challenging to classify. This evolving complexity helps address the limitations of text prompts, enhancing the overall robustness of the model.

## 5 CONCLUSION

In this paper, we present the first attempt at unifying semantic prompts and discriminative visual cues via complementary mechanisms, which is a new insight of CLIP-based model adaptation for FAS tasks. We address the challenge of generalizable face anti-spoofing (FAS) by introducing a novel framework, namely CTV-FAS, that enhances robustness against sophisticated attacks, such as high-resolution replay attacks, that are difficult to describe linguistically. In the training process, visual cues are generated from the Self-Supervised Consistency Module (SSCM) to improve the generalization capabilities of the visual anchor cache. Subsequently, visual anchors are dynamically optimized by the Visual Anchors Updating Module (VAUM), which selects hard language-insensitive samples. During inference, to effectively combine visual and textual cues, we introduce an Adaptive Modality Integration Module (AMIM), which ensures seamless fusion of both modalities, optimizing their synergy. The proposed method has been rigorously tested, demonstrating a significant improvement over existing state-of-the-art solutions in FAS tasks, as evidenced by our comprehensive experimental results and analyses.

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

## A  APPENDIX

### A.1  SUPPLEMENTARY ABLATION STUDY

Table 9: Effects of the number $n$ of the hard visual feature to update visual prompt.

| Num | C → I | | C → M | | C → O | | Avg. |
|---|---|---|---|---|---|---|---|
| | HTER | AUC | HTER | AUC | HTER | AUC | HTER |
| 10 | **11.64** | 92.03 | **1.72** | **99.27** | 2.57 | **99.73** | **5.31** |
| 30 | 12.31 | **92.48** | 2.52 | 98.65 | **2.36** | 99.56 | 5.73 |
| 50 | 12.27 | 91.83 | 1.73 | 98.91 | 3.48 | 99.31 | 5.82 |
| ALL | 14.75 | 91.08 | 3.59 | 98.81 | 4.57 | 98.94 | 7.64 |

Table 10: Effects of weight scaling degree $\alpha$.

| Num | C → I | | C → M | | C → O | | Avg. |
|---|---|---|---|---|---|---|---|
| | HTER | AUC | HTER | AUC | HTER | AUC | HTER |
| 0 | 14.22 | 90.15 | 4.11 | 98.78 | 5.44 | 98.72 | 7.92 |
| 1 | 13.54 | 90.87 | 2.27 | 99.46 | 3.39 | 99.48 | 6.4 |
| 3 | **11.64** | 92.03 | **1.72** | **99.27** | 2.57 | **99.73** | **5.31** |
| 5 | 12.31 | **92.49** | 2.64 | 99.11 | **2.33** | **99.73** | 5.76 |

**Effects of weight scaling degree $\alpha$:** Tab. 10 demonstrates the influence of the weight scaling factor on the outcomes of ensemble methods. When the scaling factor $\alpha$ is set to 0, the method is tantamount to a simple average ensemble. As the value of $\alpha$ exceeds 1, the scaling mechanism adjusts the fusion weights, amplifying the influence of components with lower entropy and diminishing the impact of those with higher entropy. The empirical results suggest that the optimal generalization performance of the model is achieved with a scaling factor of 3. Conversely, the approach yields the least effective results when $\alpha$ is 0, highlighting the limitations of average aggregation. These findings underscore the efficacy of adjusting fusion weights in enhancing the generalization capabilities of the model.

**Effects of the number $n$ of the hard feature to update visual anchor:** To thoroughly understand the impact of hard visual anchor updates on generalization performance, we explore the number of visual anchor features integrated per epoch as shown in Tab. 9. Experimental results indicate that optimal generalization performance is achieved when the top 10 +2.33 increase in average HTER) challenging samples are integrated per epoch. Conversely, incorporating all visual features results in the poorest generalization performance. This suggests that blending an appropriate amount of difficult samples into visual anchors complements semantic text prompts effectively, thus enhancing generalization performance.

