# OpenReview forum: "CTV-FAS: Compensate Texts with Visuals for Generalizable Face Anti-spoofing"
_ICLR.cc/2025/Conference — ICLR 2025 Conference Withdrawn Submission_

### Official Review · Reviewer_hsdb · 2024-10-31

**Soundness:** 3
**Presentation:** 3
**Contribution:** 2
**Rating:** 3
**Confidence:** 5

**Summary:**

This work proposes a new FAS method using vision-language pre-trianed models. In particular, since previous methods fail to model hard samples such as high-resolution image attack, this work designs several modules to deal with it. Self-Supervised Consistency Module (SSCM) is introduced to align the vision and language feature space in a teacher-student self-supervised manner. And Visual Anchors Updating Module (VAUM) is specifically designed for hard samples. During inference, Adaptive Modality Integration Module
(AMIM) is utilized to ensemble the results of visual anchors and text prompts. Extensive experiments on several popular datasets show the effectiveness of the proposed method.

**Strengths:**

The strengths are listed as below:
(1) This work is easy to follow and the idea is clearly stated.
(2) Extensive experiments are conducted. And the experimental results are good.
(3) The visualizations are good.

**Weaknesses:**

The weaknesses are listed as below:
(1) The overall framework seems simple and the contributions are marginal. For instance, Self-Supervised Consistency Module (SSCM) is a simple application of self-supervised learning frameworks. It can hardly be noticed as a contribution.
(2) The design of VAUM is intuitive. It adopts a moving average between current anchors and the hard samples. As a result, the choice of beta and the choice of hard samples can dramatically influence the final performance. There is no detailed ablation studies on the choice of beta and the choice of hard samples.
(3) According to the ablation results in Tab. 4, with the help of VAUM, the baseline is significantly improved. However, it seems that baseline+VAUM does not use visual anchors for decision making. Then, how does it happen that VAUM helps improve the performance.

**Questions:**

(1) Is the idea of visual anchors first proposed in this work?
(2) It is interesting to show the mis-classified samples after adding VAUM, which are correctly predicted without VAUM.

---

> ### Comment · Reviewer_hsdb · 2024-11-27
> **Comment by reviewer hsdb**
>
> Since there is no feedback, I keep my rating to reject the work.

---

### Official Review · Reviewer_BABT · 2024-10-31

**Soundness:** 2
**Presentation:** 2
**Contribution:** 2
**Rating:** 3
**Confidence:** 4

**Summary:**

This paper proposes a novel framework termed CTV-FAS to address the limitations of existing methods that certain attacks cannot be described linguistically. Specifically, three modules are introduced: Self-Supervised Consistency Module (SSCM) to utilize consistency regularization to facilitate visual feature learning, Visual Anchors Updating Module (VAUM) to incorporate the visual anchors through an adaptive updating scheme, Adaptive Modality Integration Module (AMIM) to merge visual and textual information during inference seam-
lessly. Experiments on public benchmarks demonstrate the effectiveness of proposed method.

**Strengths:**

* Incorporating the pretrained vision-language model for FAS tasks is good and this has been proved to be effective on many other computer vision tasks.
* The problem the authors aim to solve: some settings cannot be described by text prompts also make sense, and is also a issue for related fields.
* The experiments are strong enough to support the effectiveness of proposed method on FAS task.

**Weaknesses:**

* The introduce of visual cues: the authors aim to solve the problem (some attacks cannot be described by texts) by introducing visual anchor, but the image input is already one kind of visual cues, is there any evidence that another visual cues can actually be helpful for existing visual cues? Also the idea which tries to solve the problem of text modality by designing on visual modality seems weird intuitively.
* For Fig.1: It is clear that from (a) the generalization is affected, but it is not that clear from (b) how the generalization is improved by V1. Is it because V1 is closer to T1? This part should be clear.
* The authors mainly target on two attacks: paper and replay attacks. Is this method also be effective for other attacks such as 3D mask?
* For SSCM: which part of visual encoder is updated during training? From Fig.2 (1), the teacher part is frozen, but from Eq. (2), the parameters of teacher is updated. It is not clear. And why masking can make the feature more robust for this task? More explanations are needed.
* For VAUM: we assume the part in Fig.2 (2) is applicable, the anchor for hard samples should be highly-related to the sample in training data, how can the authors make sure the anchors in training data can generalize well to unseen distributions? More proof is needed. Besides, the CLIP encoders are not fine-tuned, how can they have the knowledge of which attack is employed (print/replay/real) ? This is also weird.
* For AMIM: the calibration by the entropy is kind of widely-used technique in related field, the novelty of this part should be justified.
* For benchmark: the authors should compare more recent methods, they only compare with one method on 2024, which should not be enough.
* For ablation: are all the condions discussed in Tab.4? How about only using SSCM and AMIM?
* For Tab.7: what do you mean by only using one branch? Is this only during inference?
* For Fig. 3: it could be better and more clear if you show how each module influences the decision boundary on source/target distributions.

**Questions:**

There are some unclear points that need to be justified. Please see the weaknesses part. I will raise the score if the authors address them properly during rebuttal.

---

### Official Review · Reviewer_Pj5j · 2024-11-01

**Soundness:** 2
**Presentation:** 2
**Contribution:** 2
**Rating:** 5
**Confidence:** 5

**Summary:**

This paper introduces the CTV-FAS framework to improve generalizable face anti-spoofing (FAS) by integrating visual anchors with text prompts to enhance domain generalization in vision-language models. The CTV-FAS comprises three key modules:
1. Self-Supervised Consistency Module (SSCM), capturing more granular representations and enhancing the model’s consistency and robustness.
2. Visual Anchors Updating Module (VAUM), compensating for the limitations of semantic prompts.
3. Adaptive Modality Integration Module (AMIM), balancing visual and textual information during inference.

Overall Assessment:
1. The explanations of the three main contributions (Sec. 3.2, 3.3, and 3.4) have limited pages, which may leave readers confused about technical details.
2. The innovation of SSCM and VAUM appears limited.

[1] Improving CLIP Training with Language Rewrites

**Strengths:**

1. Thorough Evaluation: This work conducts extensive experiments across multiple datasets, and includes thorough ablation studies, providing a comprehensive assessment.
2. State-of-the-Art Performance: Experimental results demonstrate that CTV-FAS outperforms existing FAS methods in a thorough evaluation.

**Weaknesses:**

1. Regarding the limited innovation of SSCM: SSCM doesn't seem to be an innovative technology in essence. It seems that it is just a combination of two commonly used self-supervised learning paradigms, i.e., MoCo and MAE. Additionally, it seems unreasonable to directly increase the similarity between a severely damaged (75% masked) sample and an augmented sample. Data augmentations in contrastive learning typically rely on consistent augmentation methods to ensure alignment in representation/semantic space. The core idea of contrastive learning is that the two augmented samples should have a high consistency in representation/semantics space. While MAE, which based on masked content prediction, is an implicit mutual information maximization approach. Given the high redundancy in image data, partial pixel representation is sufficient to predict the original image. It aims to learn robust/universal visual representations by learning the correlation between pixels. In addition, the author also propose to use SimCLR loss in Sec. 3.5. I am quite confused about all the settings mentioned above and hope the author can explain the insight or motivation.

2. About the semantic prompts in VAUM: The semantic prompts mentioned in your article, as shown in Figure 2 (2), e.g., ‘This is a real face’, ‘This is a print face’, ‘This is a replay face’ are simple prefix-suffix templates of text prompts that may lack sufficient semantic richness, which may result in limited performance. Some works [1] related to VLMs have already pointed out this issue and proposed using LLMs for text enhancement, rewriting, and expansion to enhance semantics. So simple prefixes or suffixes template may not truly be considered as semantic prompts, and the most direct approach may be to use text with rich semantics.

3. Insufficient explanation of VAUM: The VAUM section lacks sufficient detail, particularly regarding the selective updating strategy. In addition, how many visual anchors and text anchors are used? These aspects are not clearly specified.

**Questions:**

1. Please refer to Weakness 1.
2. The SSCM and VAUM employ the EMA update strategy, but the specific momentum coefficient of SSCM and VAUM are not provided.
3. Is the model performing fine-grained classification by labeling each type of attack, or is it a binary classification? It seems that the detailed descriptions and processing on this aspect are lacking, as some datasets may not have clearly labeled the types of attacks.

---

### Official Review · Reviewer_WeK8 · 2024-11-03

**Soundness:** 3
**Presentation:** 2
**Contribution:** 3
**Rating:** 6
**Confidence:** 4

**Summary:**

The paper presents CTV-FAS, a novel framework for Domain Generalization Face Anti-Spoofing (FAS) that aims to improve spoof detection across diverse domains by combining information from semantic prompts and visual anchors. Previous FAS methods only leveraging semantic prompts face limitations when dealing with replay attacks which is difficult to describe purely through language. CTV-FAS addresses this by incorporating visual cues, which provide complementary information to text prompts.

 The framework includes three core modules:

**Self-Supervised Consistency Module (SSCM)**, which enhances the robustness of visual anchors by enforcing consistency across various augmented views of images.

**Visual Anchors Updating Module (VAUM)**, which dynamically updates visual anchors during training to get most challenging visual features that text prompts cannot capture.

**Adaptive Modality Integration Module (AMIM)**, which fuses the predictions from text and visual anchors during inference, adjusting their weights based on prediction confidence.

The paper demonstrates that this model outperforms current state-of-the-art methods in FAS leave-one-out tasks, showing its effectiveness in enhancing model generalization and robustness against complex spoofing attacks.

**Strengths:**

1. This paper introduces a novel combination of semantic and visual anchors for face anti-spoofing, effectively addressing limitations of text-only prompts for complex attacks. This approach creatively adapts vision-language models for enhanced generalization in FAS.
2. The framework is well-designed with effective modules—SSCM, VAUM, and AMIM—each shown to improve performance.
3. The paper is clearly structured, with figures, and tables that aid in understanding the architecture and experimental results.
4. By enhancing cross-domain generalization, especially against high-resolution spoofing attacks, this work makes a meaningful contribution to FAS.

**Weaknesses:**

**Limited Anchor Diversity Cause Generalization Limitations**: The use of only three anchors (replay, print, real) may constrain the model’s ability to generalize to a wider range of open-world attacks, particularly those not represented in the training data. This limited diversity in anchors could impact the model's performance in real-world scenarios. Additionally, since the visual anchors are learned exclusively within the training domains and do not adapt to new patterns, the framework may face challenges when encountering previously unseen attack variations.

**Questions:**

**Entropy-Based Fusion**: Why did the authors choose entropy as the metric for weighting the fusion of visual and semantic anchors, rather than exploring machine learning techniques to integrate information from both anchors? Using a learned approach, such as a neural network for fusion, could potentially capture more nuanced interactions between the modalities.

 **Visual Anchor Updating Strategy**: The visual anchors are updated with in few in-domain samples, but rare or unusual attacks may still be underrepresented. How does the method ensure stability and robustness when facing rare attack types?

**Anchor Diversity and Generalization**: Given that only three types of anchors (replay, print, real) are used, how well does the model handle more complex or open world attacks? Would increasing the number or variety of anchors improve generalization to unseen attack types?

We hope the author can help resolve our confusion.

---

### Official Review · Reviewer_BMHM · 2024-11-04

**Soundness:** 3
**Presentation:** 3
**Contribution:** 3
**Rating:** 5
**Confidence:** 4

**Summary:**

CTV-FAS proposed a Vision-Language approach to solve domain generalization in Face Anti-Spoofing, but the proposed method lacks novelty and contributions to the society.
This work studies generalizable Face Anti-Spoofing. It utilized Vision-Language Models CLIP to address the domain generalization problem by using the Self-Supervised Consistency Module (SSCM), Visual Anchors Updating Module (VAUM), and Adaptive Modality Integration Module (AMIM). CTV-FAS unifies semantic prompts and discriminative visual cues via complementary mechanisms. Empirically, the model achieves the SoTA performance in the ICMO and SCW datasets.

**Strengths:**

1.	The authors conduct extensive experiments and analyses on the proposed method, effectively showing its advantages.

2.	The proposed CTV-FAS performs greatly satisfactorily on both widely used domain generalization datasets ICMO and SCW.

**Weaknesses:**

1.	This work is very similar to the work [1] with the Teacher-Student mechanism and the combination process by exchanging the image-end and the text-end. Thus, this work may not be the first one doing that and I question the novelty of this work. Maybe the authors can address the differences and clarify the situation.
2.	Although the method has achieved satisfying results on the cross-domain experiments, it is still fuzzy how to compose such modules together that can help address the domain generalization problem. Maybe the authors should discuss more about the motivation of this work and the relationship between the task and the proposed algorithm.
3.	Some works, like [2] [3], are not cited and are essential to the field. The authors should pay more attention to this.
4.	There is a lack of comparison between the baseline methods like CLIP, CoOp, CoOpOp in the experiments.

[1] Unified physical-digital face attack detection

[2] CFPL-FAS: Class Free Prompt Learning for Generalizable Face Anti-spoofing

[3] Style-conditional Prompt Token Learning for Generalizable Face Anti-spoofing

**Questions:**

See the disadvantages above.

---

### Official Review · Reviewer_4saS · 2024-11-04

**Soundness:** 4
**Presentation:** 3
**Contribution:** 3
**Rating:** 5
**Confidence:** 5

**Summary:**

This paper proposes to use  vision-language model for Face Anti-Spoofing, focusing on the text prompt. This  paper found out  certain attacks, such as high-resolution replay attacks, cannot be described linguistically.  To tackle this limitation, this paper proposes to exploit visual anchors to compensate for the shortcomings of semantic prompts.  SSCM is proposed to  boost the generalization of visual anchors. VAUM is proposed to incorporate the visual anchors through an adaptive updating scheme. AMIM is designed to merge visual and textual information during inference seamlessly. However,  there are some  presentation  issues. I give "boarderline" decision at this round. After rebuttal and discussion with other reviewers, I give my final rating.

**Strengths:**

The motivation is justified that the semantic prompt is hard to describe high-resolution replay attacks linguistically.
The proposed method  achieves  state-of-the-art performance. Overall, the description  of the method is clear. However, there are some issues  of  this paper.

**Weaknesses:**

The SELF-SUPERVISED CONSISTENCY MODULE is a self-distillation technique, and its relathionship and  motivation for the visual anchor  is unclear. It looks like that this paper simply combines the SSCM with  other modules (VAUM and  AMIM )


There are some  writing and  presentation issues, including citation format. For example, "mean-weighted Ming & Li (2024)" should be "mean-weighted (Ming & Li, 2024)".

In the  Introduction, this paper raised two questions "What visual cues are robust enough to differentiate between the real
person and paper/replay attack? ",  and "What visual anchors can compensate  for the deficiency of semantic prompts". How is unclear how the questions  are answered. It is suggested that the paper  highlight the answers of these two questions somewhere in the paper.

Fig.3: the color used in the visualization is not discriminate. The advantage of the proposed method  is unclear as shown in the Fig.3.

**Questions:**

N..A.

**Details Of Ethics Concerns:**

N.A.

---

### Official Review · Reviewer_Duev · 2024-11-05

**Soundness:** 3
**Presentation:** 3
**Contribution:** 3
**Rating:** 6
**Confidence:** 4

**Summary:**

The paper presents the CTV-FAS framework, a new approach in generalizable Face Anti-Spoofing (FAS) that integrates both visual and semantic cues to improve robustness against sophisticated attacks. Unlike previous methods that rely solely on semantic prompts, CTV-FAS uses visual anchors to address limitations in representing complex spoofing attacks. Key components of this framework include the Self-Supervised Consistency Module (SSCM), Visual Anchors Updating Module (VAUM), and Adaptive Modality Integration Module (AMIM), which work in tandem to ensure robust performance across domains.

**Strengths:**

1. The paper is well-written. The proposed face anti-spoofing model is constrained to focus on local detail information, with hard samples selected to aid training.

2. The experimental results indicate a significant performance improvement.

**Weaknesses:**

The dynamic visual anchor updating may result in bias toward visual characteristics seen during training.

**Questions:**

1. Using image embeddings of hard samples as supervisory information may lead to overfitting on the training samples. How is the generalization ability for unseen spoofing patterns of the same category?

2. Could this approach affect the judgment of simpler samples? Sometimes, the spoofing patterns of hard samples differ from those of simpler ones.

3. In Table 4, what method is used as the baseline? What is the impact of SimCLR on performance?

4. Based on Tables 3 and 7, the performance of the semantic prompt branch is also notable. Could it be that CLIP’s understanding of categories (e.g., print attacks) already covers the spoofing patterns of category-related fake faces, even if we consider it challenging to describe these spoofing patterns in language?

5. As seen in Table 7, the performance of the visual anchor and semantic prompt branches is comparable. After integration, the performance improves. Could this improvement result from ensemble learning, rather than visual anchors specifically addressing issues semantic prompts cannot?

6. Based on Tables 7 and 8, it seems that only certain ensemble methods yield performance gains. Simple averaging does not suffice; a carefully designed integration is necessary.

7. Figure 1 is somewhat difficult to understand; it’s hard to see the impact of adding visual anchors.

---

### Note · Authors · 2024-11-27

I have read and agree with the venue's withdrawal policy on behalf of myself and my co-authors.